# Single-Stage Visual Query Localization in Egocentric Videos

**Hanwen Jiang**[1]  **Santhosh Kumar Ramakrishnan**[1]  **Kristen Grauman**[1,2]

UT Austin[1]    FAIR, Meta[2]

## Abstract

Visual Query Localization on long-form egocentric videos requires spatio-temporal search and localization of visually specified objects and is vital to build episodic memory systems. Prior work develops complex multi-stage pipelines that leverage well-established object detection and tracking methods to perform VQL. However, each stage is independently trained, and the complexity of the pipeline results in slow inference speeds. We propose *VQLoC*, a novel single-stage VQL framework that is end-to-end trainable. Our key idea is to first build a holistic understanding of the query-video relationship and then perform spatio-temporal localization in a single shot manner. Specifically, we establish the query-video relationship by jointly considering *query-to-frame* correspondences between the query and each video frame and *frame-to-frame* correspondences between nearby video frames. Our experiments demonstrate that our approach outperforms prior VQL methods by 20% accuracy while obtaining a 10× improvement in inference speed. VQLoC is also the top entry on the Ego4D VQ2D challenge leaderboard. Project page: https://hwjiang1510.github.io/VQLoC/

## 1  Introduction

Episodic memory, a specific type of long-term memory in humans, facilitates the retrieval of our past experiences such as their time, location, related context, and activities [46]. This form of memory is pivotal to the progressive evolution of embodied intelligence due to its foundational role in enabling long-horizon reasoning [4, 19]. Nevertheless, human memory can falter in accurately recalling the details of daily activities—such as where we left our keys, or who we met while going for a run. This perceptual overload can be mitigated through the utilization of head-worn cameras or AR glasses to capture our past experiences and the development of systems that can retrieve information. To this end, the episodic memory benchmark [13] is recently proposed, which aims to develop systems that enable super-human memory by allowing humans to query about their past experiences recorded on head-worn devices. These systems also have the potential to serve as a conduit for learning from past experiences and data [60, 44], promoting long-term robot learning [25].

We focus on 2D *Visual Query Localization* (VQL), a vital task in the episodic memory benchmark that aims to answer the question *"where did I last see object X?"*. Specifically, given a long-form egocentric video representing past experiences from a human camera-wearer, the goal is to *search and localize* a visual query object, specified as a visual image crop, in space-time (see Fig. 1, left). Addressing VQL is of paramount importance, as query object localization serves as a prerequisite for downstream object-centric reasoning tasks.

VQL presents a myriad of challenges to existing computer vision systems. For example, current object detection systems localize pre-defined object categories on well-curated internet-style images [40, 5]. In contrast, VQL requires localizing an *open-set* of objects specified as visual queries. Current object tracking systems expect to be initialized with a bounding box of the object next to its appearance in the video [54, 31]. However, the visual query crop of VQL originates from an image *outside*

37th Conference on Neural Information Processing Systems (NeurIPS 2023).

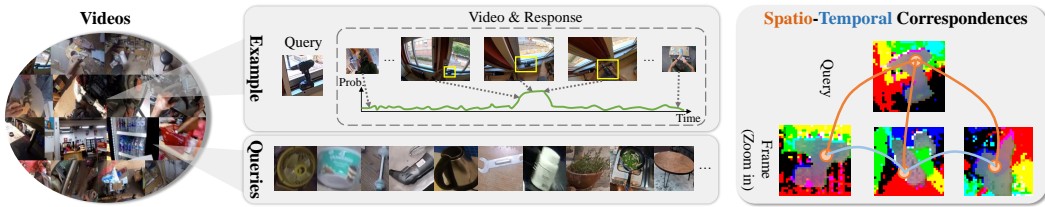

Figure 1: **Visual Query Localization (VQL)**: (Left) The goal is to localize a visual query object within a long-form video, as demonstrated by the response track marked by yellow bounding boxes. The complexity of this task arises from the need to accommodate an open-set of object queries with diverse scales, viewpoints, and states as they appear in the video. (Right) Our method VQLoC first establishes a holistic understanding of the query-video relationship by jointly reasoning about query-to-frame (spatio) and frame-to-frame (temporal) correspondences, before localizing the response in a single-stage and end-to-end trainable manner.

of the video being queried; that is, there may not be an exact match and "close frame" for the visual query. Complicating matters further, the egocentric nature of the VQL task presents its own unique challenges. Compared with the visual query, the object may appear in the video under significantly varying orientations, sizes, contexts, and lighting conditions, and it may experience motion blur and occlusions. Finally, egocentric videos can span several minutes, hours, or days in real-world applications, while the object itself may appear only for a few seconds, resulting in a "needle-in-the-haystack" problem.

Prior work has attempted to address VQL through a bottom-up framework with three stages [13, 52, 53]: i) in each video frame, detect all objects and conduct pairwise comparisons with the visual query to obtain the proposal that is most similar to the query; ii) identify the similarity score peaks throughout the video; and iii) perform bidirectional tracking around the most recent peak to recover the spatio-temporal response. Although this framework is grounded in well-established object detection and tracking approaches, it relies heavily on the first stage to detect the object by *independently looking at each frame*. While this may be possible if the object appears clearly in the video, it is often not the case due to the egocentric nature of images. Errors in frame-level object detection can potentially cause the entire system to fail since the framework is not end-to-end differentiable and errors in earlier stages may not be rectifiable in later stages. Moreover, the methods suffer from a slow inference speed due to the high complexity of pairwise comparison with redundant object proposals.

To address these limitations, we propose VQLoC (Visual Query Localization from Correspondence), a novel single-stage framework for VQL. Our key insight in VQLoC is that a holistic understanding of the query-video relationship is essential to reliably localize query objects in egocentric videos. Accordingly, VQLoC jointly models the *query-to-frame* relationship between the query and each video frame and *frame-to-frame* relationships across nearby video frames (see Fig. 1, right), and then performs spatio-temporal localization in a single-stage and end-to-end trainable manner. Specifically, we establish the query-to-frame relationship by extracting image features for the visual query and each video frame using a ViT backbone pretrained with DINO [37] and using a cross-attention transformer module [48] to establish correspondences between the image regions in the query and video frame. We then propagate these correspondences over time using a self-attention transformer module [48] that exploits the frame-to-frame relationships resulting from the temporal continuity of videos to capture the overall query-video relationship. Finally, we use a convolutional prediction head that performs spatio-temporal localization by utilizing the query-video relationship to make frame-level predictions. Critically, our model operates in a single stage, i.e., there are no intermediate localization outputs with dedicated post-processing steps, and it is end-to-end trainable since it uses only differentiable modules to obtain the final prediction.

VQLoC has several benefits when compared with the prior stage-wise methods. Unlike prior work that explicitly generates object proposals in the video frame and compares them with the visual query, VQLoC implicitly establishes the query-frame relationship by performing attention-based reasoning between the visual query features and the video frame features. This approach effectively utilizes image regions of the background and non-query objects as contextual information for inference. Additionally, our implicit query-frame relationships are significantly faster to compute than explicitly generating proposals and performing pairwise comparisons, which is critical for real-world episodic memory applications. Finally, VQLoC is end-to-end trainable, which results in much better performance compared to prior work.

We evaluate VQLoC on the Ego4D Visual Query 2D Localization benchmark. VQLoC achieves the state-of-the-art on this benchmark, outperforming previous methods by $20\%$, and it is also ranked first on the public Ego4D VQ2D challenge leaderboard.[1] Furthermore, VQLoC achieves real-time inference with 36 Frames Per Second (FPS), which is $10\times$ faster than prior work. We are committed to releasing our code upon publication for future research.

## 2 Related Work

**Object detection.** Prior work has made impressive progress in building state-of-the-art object detectors for localizing objects in images [26, 12, 40, 5]. These methods typically detect objects with pre-specified object categories and are evaluated on internet-style and human-curated image datasets [27, 10, 56, 38]. While these methods relied significantly on having sufficient training data, recent work has extended them to incorporate long-tail object categories [14, 43] and free-form language-based detection [58]. Prior work has also proposed methods for one-shot detection, which build on the success of object detection methods [17, 34, 7, 57]. They leverage objectness priors by calculating similarities between visual queries and region proposals. However, all of the methods work on well-curated images, e.g. the COCO dataset [27]. Our work focuses on the VQL task, which is a challenging version of one-shot detection on long-form egocentric videos. Unlike image-based one-shot detection where the object is assumed to be present in the image, video-based one-shot detection requires us to identify *in which frame* the object appears for spatio-temporal localization.

**Object Tracking.** There is a rich literature on object tracking methods that track a static or dynamic object throughout a given video [8, 33, 54, 31, 22, 36, 35]. The object is typically specified as a bounding box in a video frame, and the goal is to track it through the subsequent frames in the same video [24, 11]. Short-term tracking methods are developed to track objects till they completely disappear from the video frame [47, 30], while long-term tracking methods are capable of recovering the object track when it re-appears in the video [50, 32, 54]. Tracking methods are initialized with an object bounding box close to its appearance in the video, while visual queries in VQL may originate from outside the video. This can pose a challenge for tracking methods. We compare with a state-of-the-art long-term tracker in our experiments [54]. Technically, VQLoC differs from tracking-based methods in using a spatial-aware transformer, temporal reasoning within local windows, and anchor-based prediction heads.

**Visual Query Localization (VQL).** The recently proposed episodic memory benchmark introduces the VQL task on egocentric videos, where the goal is to spatio-temporally localize a query object specified as a visual image crop [13]. Prior work has tackled VQL using a three-stage detection and tracking framework. First introduced as baseline in Ego4D, this approach performs frame-level detection, identifies the most recent "peak" in the detections across time, and performs bidirectional tracking [3] to recover the complete temporal extent of the response [13]. Subsequent research in this paradigm improves the frame-level detector by reducing false positive predictions through negative frame sampling [52] and proposes using background objects as context [53]. However, these multi-stage approaches independently perform spatial and temporal reasoning in a modular fashion without end-to-end training, which results in compounding errors across the stages. Furthermore, they incur significant computational costs and time for training and inference, which limits their applicability to real-time episodic memory applications. Our proposed method, VQLoC, distinguishes itself by first establishing holistic spatio-temporal correspondences and then localizing the object with an end-to-end paradigm. This leads to state-of-the-art results with fast inference speeds.

**Visual Correspondence.** Establishing correspondence is a core problem in computer vision [15]. The advancement of visual correspondence models has significantly contributed to the success of various computer vision tasks, including image matching and retrieval [45, 42], tracking and flow estimation [2, 16], 3D vision [29, 21], and representation learning [51, 49]. Recent developments in vision foundation models have demonstrated remarkable capabilities to establish correspondence between multi-modal [39] and spatio-temporal [6] inputs, empowering new applications [23, 20].

The design of model architectures plays a critical role in finding correspondence. Specifically, transformer models [48, 1, 59], which propagate information by establishing similarity-based correspondence, are widely used. For example, TubeDETR [55], which performs visual question answering, concatenates the text query and frame features and finds their correspondence using self-attention transformers. STARK [54] employs a similar architecture and extends it to visual templates for visual object tracking. These methods do not differentiate the query features and the

---

[1]Ego4D VQ2D challenge leaderboard: https://eval.ai/web/challenges/challenge-page/1843/leaderboard

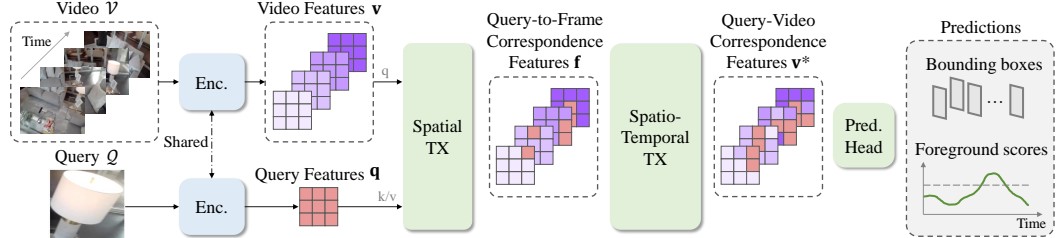

Figure 2: **Visual Query Localization from Correspondence (VQLoC)**: We address the VQL problem by first establishing the overall query-video relationship and then inferring the response to the query. VQLoC extracts image features for both the query and video frames independently. It establishes query-to-frame correspondence between the query features $\mathbf{q}$ and each video frame feature $\mathbf{v}_i$ using a transformer module (Spatial TX), and obtains the correspondence features $\mathbf{f}$ for identifying potential query-relevant regions in the frames. VQLoC then propagates these frame-level correspondences through time using another transformer module (Spatio-Temporal TX), which leverages the frame-to-frame correspondence between nearby video frames and obtains the query-video correspondence features $\mathbf{v}^*$. Finally, VQLoC uses the query-video correspondence to predict the per-frame bounding box and the probability that the query object occurs in the frame.

frame features after the concatenation, making the features lose spatial correlation with the input frames. In contrast, VQLoC utilizes the cross-attention transformer to establish the query-to-frame correspondence, updating the frame features by incorporating query features. This maintains the spatial correlation between frame-level features and the frames, providing strong priors for object localization.

## 3 Method

We propose a novel end-to-end framework for localizing visual queries in egocentric videos. We design our model based on the insight that a holistic understanding of the query-video relationship is crucial for reliably localizing query objects appearing in the video. To this end, we propose to first capture both the *query-to-frame* relationship for each video frame and the *frame-to-frame* relationship across nearby frames. We then predict the results based on the captured query-video relationship in a single shot. Next, we will describe the formulation of the VQL task in Sec. 3.1, introduce our model architecture in Sec. 3.2, and describe our approach for training and inference in Sec. 3.3.

### 3.1 VQL Task Formulation

The VQL task is formulated as open-set localization in egocentric videos, answering questions like "where did I last see my wallet?". Formally, given an egocentric video $\mathcal{V} = \{v_1, \cdots, v_T\}$ consisting of $T$ frames and a visual query $\mathcal{Q}$ specified as an image crop of the query object[2], the goal is to spatio-temporally localize the *most-recent occurrence* of the query object in the video. The localization result is a response track $\mathcal{R} = \{r_s, r_{s+1}, ..., r_e\}$, where $s$ and $e$ are start and end frame indices, respectively, and $r_i$ is a bounding box containing the query object on the $i^{th}$ frame.

### 3.2 VQLoC Architecture

Our proposed method VQLoC is illustrated in Fig. 2. VQLoC employs a shared image encoder to extract visual features for both the video frames and the visual query. We utilize the DINO [6] pre-trained ViT [9], which can capture object-level semantic correspondence under varying camera viewpoints [18]. First, VQLoC leverages a spatial transformer to find the query-to-frame correspondence between each video frame and the visual query. The spatial transformer identifies the regions in the frames that are similar to the query in semantics, handling the potential large appearance variation. VQLoC then uses a spatio-temporal transformer that propagates the query-to-frame correspondence over time by leveraging the frame-to-frame correspondence arising from temporal continuity in the video, and establishes the query-video correspondence. This process is performed within a local temporal window, handling the fast head motion of egocentric videos. Finally, it uses the holistic understanding of the query-video relationship to predict the response track. The response prediction

---

[2]The image crop of the query object is sampled from outside the ground-truth response track.

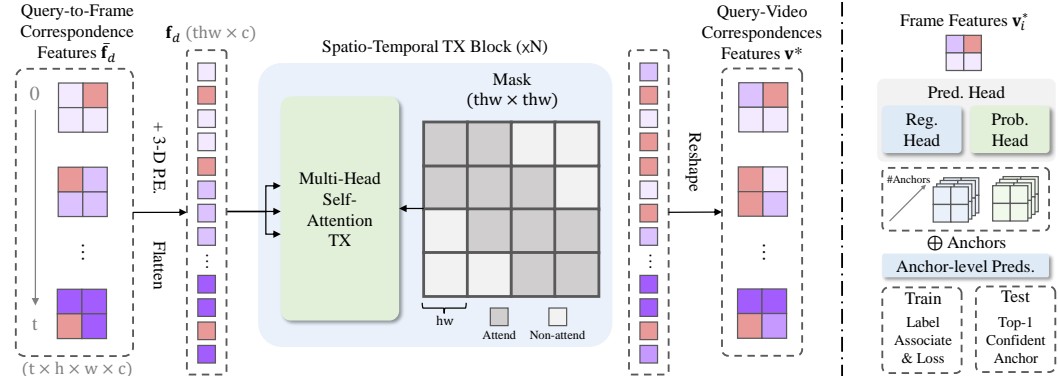

Figure 3: **Architecture for spatio-temporal transformer (left)**: We add the downsampled query-to-frame correspondence features $\bar{\mathbf{f}}_d$ with 3-D spatio-temporal positional embedding, and flatten it into 1-D tokens ($\mathbf{f}_d$). We use several transformer layers to propagate the frame-level correspondences over time by leveraging frame-to-frame correspondences between nearby video frames within a local temporal window, and obtain the overall query-video correspondence features $\mathbf{v}^*$. **Architecture for the prediction head (right)**: After establishing the query-video relationship, we use it to predict anchor-level results for each video frame $i$. We perform label association during training and pick the refined anchor with the highest probability as the prediction for the frame in testing.

head is an anchor-based model, which deals with the small scales of query objects in the video. Next, we describe these components in detail.

**Visual Encoder.** Our visual encoder consists of a fixed DINO pre-trained ViT followed by several learnable convolution layers. We independently extract visual features for each of the $T$ video frames and the visual query and obtain the video features $\mathbf{v} \in \mathbb{R}^{T \times H \times W \times C}$ and query features $\mathbf{q} \in \mathbb{R}^{H \times W \times C}$. $H, W$ signify the spatial resolution of the features, and $C$ is the channel size.

**Spatial Transformer.** The spatial transformer establishes query-to-frame correspondences between the visual crop and each video frame using cross-attention [48] between their visual features. Formally, let $v_i \in \mathbb{R}^{HW \times C}$ be the flattened video feature $\mathbf{v}_i$ for video frame $i$ and let $q \in \mathbb{R}^{HW \times C}$ be the flattened query feature $\mathbf{q}$. The query-to-frame correspondence feature $f_i$ is obtained as follows:

$$f_i = \text{FFN}(\text{CrossAttention}(v_i, q)), \tag{1}$$

where FFN is a feed-forward neural network. Intuitively, the frame features $v_i$ are updated by incorporating the corresponding query features $q$, where the cross-attention computes their feature similarity as the fusion weights. The transformer output $f_i$ is reshaped to $\mathbb{R}^{H \times W \times C}$, denoted as $\mathbf{f}_i$. Here, the updated features $\mathbf{f}_i$ retain the spatial arrangement and maintain the spatial cues for performing accurate localization. This process is repeated for all video frames to obtain the final query-to-frame correspondence features $\mathbf{f} \in \mathbb{R}^{T \times H \times W \times C}$.

**Spatio-Temporal Transformer.** The spatio-temporal transformer is designed to propagate and refine the query-to-frame correspondence features $\mathbf{f}$ from Eqn. 1 over time, by establishing frame-to-frame correspondences between nearby video frames. It leverages the temporal continuity of videos. We illustrate the architecture in Fig. 3. We first decrease the spatial resolution of the feature maps $\mathbf{f}$ using shared convolution layers, where the output is $\bar{\mathbf{f}}_d \in \mathbb{R}^{T \times h \times w \times c}$, $h, w$ and $c$ are spatial resolutions and channel size of the downsampled feature maps. We add 3-D spatio-temporal positional embedding $\mathbf{p} \in \mathbb{R}^{T \times h \times w \times c}$ to $\bar{\mathbf{f}}_d$, and flatten it into 1-D tokens $\mathbf{f}_d$:

$$\mathbf{f}_d = \text{Flatten}(\bar{\mathbf{f}}_d + \mathbf{p}) \in \mathbb{R}^{Thw \times c}, \tag{2}$$

where the positional embedding $\mathbf{p}$ is learnable and initialized as zeros.

Next, we use the spatio-temporal transformer module to establish the query-video relationship. It consists of multiple layers of locally-windowed temporal self-attention, i.e., we restrict the feature propagation within a local time window. In particular, a feature element belonging to time $t$ only attends to feature elements from time steps in the range $[t-w, t+w]$. $t-w$ and $t+w$ are the start and end of the local time window with temporal length $2w+1$. We found that this was beneficial since the locations of query objects in nearby frames are highly correlated, providing strong priors for feature refinement. We achieve locally windowed attention via masking attention weights. After performing

feature refinement using the spatio-temporal transformer module, we reshape the 1-D tokens back into the original 3-D shape to get the query-video correspondence features $\mathbf{v}^* \in \mathbb{R}^{T \times h \times w \times c}$.

**Prediction Head.** Instead of directly predicting a single bounding box for an image, we first define anchor boxes $\mathcal{B} \in \mathbb{R}^{h \times w \times n \times 4}$ on the feature maps [26]. We found that this was advantageous because the sizes and locations of query objects are diverse. The multi-scale anchor boxes can provide a strong prior on query object sizes and locations. For each frame, we define $n$ anchor boxes for each of the spatial elements of the feature map ($h \cdot w$ elements in total). Each anchor box defined on the feature map space can be mapped back to its corresponding location in the pixel space for inferring predictions. For each frame from $V_i$, we obtain the predictions by passing the query-video correspondence feature $\mathbf{v}_i^*$ through two heads:

$$\hat{\mathcal{P}}_i = \text{ProbHead}(\mathbf{v}_i^*) \in \mathbb{R}^{h \times w \times n} \quad \text{and} \quad \Delta \hat{b}_i = \text{RegHead}(\mathbf{v}_i^*) \in \mathbb{R}^{h \times w \times 4n}, \tag{3}$$

where each head consists of several convolution blocks. $\hat{\mathcal{P}}_i$ is the anchor-level query object occurrence probability and $\Delta \hat{b}_i$ is the regressed bounding box refinement. We then reshape $\Delta \hat{b}_i$ to $\mathbb{R}^{h \times w \times n \times 4}$, denoted as $\Delta \hat{\mathcal{B}}_i$. The final refined anchor boxes for the $i^{th}$ frame are $\hat{\mathcal{B}}_i = \mathcal{B} + \Delta \hat{\mathcal{B}}_i$. We perform the same operation on all frames.

### 3.3 Training and Inference

In both training and inference, we chunk the untrimmed video into fixed-length clips. During training, we ensure that the clip covers at least a part of the response track. During testing, we concatenate the predictions on all clips as the final result. We introduce the training and inference details as follows.

**Training.** VQLoC is trained with the loss $L_{img} = L_{bbox} + \lambda_p \cdot L_{prob}$ on all clip frames for supervising bounding box regression and query object occurrence probability prediction. For each refined anchor box $\hat{\mathbf{b}} \in \hat{\mathcal{B}}$, we calculate the intersection of union (IoU) between the ground-truth query object box and the original corresponding anchor box $\mathbf{b}$. If the IoU is higher than a threshold $\theta$, it will be assigned as a positive box, denoting the anchor box region captures the query object well. We apply the bounding box loss as $L_{bbox}(\hat{\mathcal{B}}, \mathbf{b}) = \Sigma_{\hat{b} \in \hat{\mathcal{B}}^p} L_{reg}(\hat{\mathbf{b}}, \mathbf{b})$, where $\mathbf{b}$ is the ground-truth box for the query object, and $\hat{\mathcal{B}}^p \subseteq \hat{\mathcal{B}}$ is the set of positive boxes. Specifically, $L_{reg}$ is a combination of the $L_1$ loss and the generalized IoU (GIoU) loss [41]:

$$L_{reg}(\hat{\mathbf{b}}, \mathbf{b}) = ||\hat{\mathbf{b}}_{\mathbf{c}} - \mathbf{b}_{\mathbf{c}}|| + ||\hat{\mathbf{b}}_{\mathbf{h}} - \mathbf{b}_{\mathbf{h}}|| + ||\hat{\mathbf{b}}_{\mathbf{w}} - \mathbf{b}_{\mathbf{w}}|| + \lambda_{giou} \cdot L_{giou}(\hat{\mathbf{b}}, \mathbf{b}),$$

where $\mathbf{b}_{\mathbf{c}}$, $\mathbf{b}_{\mathbf{h}}$ and $\mathbf{b}_{\mathbf{w}}$ are the center coordinate, height, and width of the bounding box, and $\lambda_{giou}$ is a weight that balances the losses [5]. To get the query object probability loss, we assign the probability labels to the positive and negative anchor boxes with 1 and 0, respectively. We denote the assigned probability labels as $\mathcal{P}$. The query object occurrence probability loss is defined as $L_{prob} = L_{bce}(\hat{\mathcal{P}}, \mathcal{P})$, which is the binary cross-entropy (BCE) loss.

Since the target videos are long, false positive prediction, where the model wrongly identifies other objects as the query object, is one of the bottlenecks that limit the temporal accuracy of the model. To prevent this, we perform hard negative mining (HNM) (similar to [13, 53, 52]). Given a mini-batch of $N$ videos, we diversify the negative anchors for each batch element $n$ by treating frames from every other video $n' \neq n$ as a negative. After expanding the pool of negatives, we calculate the query occurrence probability for all the negative anchors using Eqn. 3 and sample the top K anchors with the largest BCE loss as hard negatives (i.e., negatives which our model thinks are positives). We then train the model using the positives from the same batch element and the hard negatives sampled across all batch elements with the BCE loss. We keep the ratio between positives and negatives as $1 : 3$ during training.

**Inference.** During inference, we get the prediction for each frame by choosing the refined anchor box with the highest query object probability. We reject low-confidence predictions by applying a threshold $\varphi$ to the predicted query object probability. See supplementary for more details.

## 4 Experiments

First, we describe our implementation details, the dataset and evaluation metrics. We then quantitatively compare VQLoC with prior methods, and present ablation studies that examine the impact of various design choices in VQLoC.

Table 1: **VQL results on Ego4D VQ2D benchmark**: We compare our method VQLoC against the baselines. The test results are obtained from the challenge leaderboard.

| | Validation | | | | Test | | |
|---|---|---|---|---|---|---|---|
| | $tAP_{25}$ | $stAP_{25}$ | rec% | Succ. | $tAP_{25}$ | $stAP_{25}$ | FPS |
| SiamRCNN [13] | 0.22 | 0.15 | 32.92 | 43.24 | 0.20 | 0.13 | 3 |
| NFM [52] | 0.26 | 0.19 | 37.88 | 47.90 | 0.24 | 0.17 | 3 |
| CocoFormer [53] | 0.26 | 0.19 | 37.67 | 47.68 | 0.25 | 0.18 | 3 |
| STARK [54] | 0.10 | 0.04 | 12.41 | 18.70 | - | - | 33 |
| VQLoC (ours) | **0.31** | **0.22** | **47.05** | **55.89** | **0.32** | **0.24** | **36** |

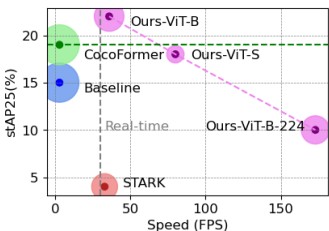

Figure 4: **Accuracy-speed trade-off**: The radius of circles notify model sizes.

## 4.1 Experimental Setup

**Implementation Details.** We train the model with $448 \times 448$ resolution videos obtained by resizing the source videos along the longer edge and applying zero-padding to the shorter edge for each frame. We set the loss coefficients $\lambda_p = 1$ and $\lambda_{giou} = 0.3$. The positive anchor box threshold is $\theta = 0.2$ due to the small scales of query objects in videos.. VQLoC is trained for $60,000$ iterations with batch size 24, utilizing the AdamW optimizer [28] with a peak learning rate of $0.0001$ and a weight decay of $0.05$. A linear learning rate scheduler and a warmup of $1,000$ iterations are employed. We adopt DINOv2's ViT-B14 as the backbone and train on clips of 30 frames with a frame rate of 5 fps. The clip length is selected to keep the balance between positive and negative frames. Clips are randomly selected to cover at least a portion of the response track. We define anchor boxes on the feature map with spatial resolution of $16 \times 16$, each element has 12 anchor boxes with 4 base sizes $(16^2, 32^2, 64^2, 128^2)$ and 3 aspect ratios $(0.5, 1, 2)$. During training, we apply random data augmentations to the clips, such as color jittering, horizontal flipping, and random resized cropping. We train and evaluate on RTX 6000 GPUs.

**Dataset.** We train and evaluate our model on the Ego4D dataset [13]. Ego4D is a large-scale egocentric video dataset designed for first-person video understanding and contains open-world recordings of day-to-day activities performed in diverse locations. We use the VQ2D task annotations from the episodic memory benchmark in Ego4D, which consists of $13.6k/4.5k/4.4k$ queries annotated over $262/87/84$ hours of train / val / test videos. The average target video duration is $\sim 140$ seconds and the average response track length is $\sim 3$ seconds, making this a challenging benchmark for the VQL task. Ego4D is, to the best of our knowledge, the only publicly available dataset for VQL.

**Metrics.** In our evaluation, we adhere to the official metrics defined by the Ego4D VQ2D benchmark. We report two average precision metrics: *temporal AP* ($tAP_{25}$) and *spatio-temporal* AP ($stAP_{25}$), which assess the accuracy of the predicted temporal and spatio-temporal extents of response tracks in comparison to the ground-truth using an Intersection over Union (IoU) threshold of 0.25. Additionally, we report the *Recovery* (rec%) metric, which quantifies the percentage of predicted frames where the bounding box achieves at least 0.5 IoU with the ground-truth. Lastly, we report the *Success* metric, which measures whether the IoU between the prediction and the ground-truth exceeds 0.05. Apart from VQ2D evaluation metrics, we also report the inference FPS of each method as a measure of computational efficiency.

**Baselines.** We compare our approach against the following baselines. All baselines are trained on the Ego4D dataset for a fair comparison.

• **SiamRCNN** [13]: This is the official baseline for VQL introduced in Ego4D. It employs a three-stage pipeline with frame-level detection [50], latest peak detection, and object tracking [3].

• **NFM** [52]: It employs the three-stage pipeline introduced in SiamRCNN and improves the frame-level detector training by sampling background frames as negative to reduce false positives. This was the winning entry of the Ego4D VQ2D challenge held at CVPR 2022.

• **CocoFormer** [53]: It improves the frame-level detection architecture from SiamRCNN by reasoning about all object proposals in a frame jointly (as a set) using a transformer module.

• **STARK** [54]: It is an end-to-end method for long-term visual object tracking. We adapt the method with our object occurrence prediction head to enable it to work on long-form videos.

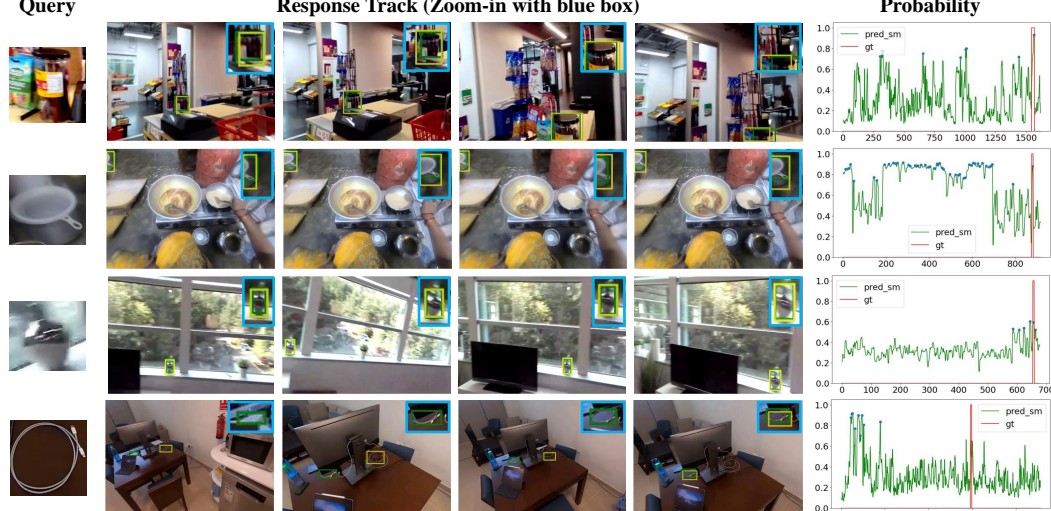

Figure 5: **Qualitative analysis of VQLoC**: In each row, we visualize an example prediction from VQLoC. For each example, we show the visual query, four frames from the predicted response track, and a temporal plot of the predicted object-occurrence probabilities (y-axis) for each video frame (x-axis). We highlight the ground-truth and predicted boxes in green and yellow, respectively. We also provide the zoom-in view of the predicted response on the top-right corner of each frame. The temporal plot additionally shows the detected peaks in blue and the ground-truth response window (which is only annotated for the latest appearance of query objects) in red. VQLoC is able to localize query objects under challenging conditions such as clutter (row 1), partial-to-near-complete occlusions (row 2), and motion blur with significant viewpoint variations (row 3). We show a failure case in row 4 where VQLoC fails to distinguish between two cables with similar appearances. Note that the temporal plot in the last column shows the object occurrence probability on the entire video, while the ground-truth is only annotated for the last occurrence.

## 4.2 Experimental results

The results are shown in Table 1. Our method VQLoC outperforms prior works on all metrics and sets the state-of-the-art for this task. VQLoC relatively improves over the next-best baseline (CocoFormer) by 19% $tAP_{25}$, 16% $stAP_{25}$, 25% recovery, and 17% success. Furthermore, the running speed of VQLoC is $10\times$ faster than the three-stage VQL methods. Besides, we also demonstrate that directly applying tracking methods, i.e. STARK, doesn't work as well for VQL. We conjecture the reason is that STARK concatenates the query features and frame features without differentiating them. In this manner, the concatenated features, which are used to regress the localization results, lose the spatial correlation with input frames. While this design might work for normal videos where the camera viewpoint changes slowly and smoothly, its problem is exacerbated in egocentric videos where the camera moves dramatically and the object specified by the visual query is quite different from its appearance in the video. As shown in Fig. 4, VQLoC shows a reasonably good balance on the speed-accuracy trade-off. At 80 FPS, we match the best baseline performance; at 33 FPS, we outperform the best baseline by a healthy margin. We present a qualitative analysis in Fig. 5.

## 4.3 Ablation Study

We present an ablation study on each block of VQLoC, examining and evaluating their impact within the structure of our model.

**Backbone Size and Input Resolution.** We explore the impact of using different backbone sizes and input spatial resolutions. We test the following alternatives: i) using a smaller ViT-s14 backbone with $448 \times 448$ resolution, and ii) using the original ViT-b14 backbone but a smaller $224 \times 224$ resolution. Specifically, the input with spatial resolutions $448 \times 448$ and $224 \times 224$ result in features with spatial size $32^2$ and $16^2$, respectively. As shown in Table 2, the model with the largest input size and the largest backbone achieves better results. However, experiments demonstrate that using a larger backbone only brings marginal gains (comparison between the last two rows). In contrast, when inputting the model with low resolution, the performance drops dramatically. Specifically, the ratio of $tAP_{25} : stAP_{25}$ drops from 66% to 45%. This decrease reflects the huge degeneration of

Table 2: Ablation on the backbone size choices. Res. denotes the spatial resolution of the inputs.

| Backbone | Res. | tAP$_{25}$ | stAP$_{25}$ | rec% | Succ. |
|---|---|---|---|---|---|
| ViT-b14 | 224 | 0.22 | 0.10 | 32.18 | 45.84 |
| ViT-s14 | 448 | 0.28 | 0.18 | 37.71 | 52.18 |
| ViT-b14 | 448 | **0.29** | **0.19** | **38.58** | **53.12** |

Table 3: Ablation of finding frame-query correspondence. X-TX denotes our spatial transformer.

| Fusion | Layers | tAP$_{25}$ | stAP$_{25}$ | rec% | Succ. |
|---|---|---|---|---|---|
| Conv | - | 0.22 | 0.13 | 32.27 | 44.95 |
| X-TX | 1 | **0.29** | 0.19 | **38.58** | 53.12 |
| X-TX | 3 | **0.29** | **0.20** | 37.91 | **53.74** |

Table 4: Ablation of spatio-temporal transformer. Win. is the window size, and glb. means global.

| ST Refine. | Win. | Layers | tAP$_{25}$ | stAP$_{25}$ | rec% | Succ. |
|---|---|---|---|---|---|---|
| ST-TX | glb. | 3 | 0.08 | 0.03 | 12.49 | 15.73 |
| ST-TX | 9 | 3 | 0.27 | 0.18 | 37.98 | 51.08 |
| ST-TX | 7 | 3 | 0.27 | **0.19** | 36.97 | 50.21 |
| ST-TX | 5 | 3 | **0.29** | **0.19** | **38.58** | **53.12** |

Table 5: Ablation on the prediction head and loss function. FL denotes focal loss.

| Anchor | Loss | tAP$_{25}$ | stAP$_{25}$ | rec% | Succ. |
|---|---|---|---|---|---|
| $\times$ | BCE | 0.17 | 0.10 | 34.23 | 47.60 |
| $\checkmark$ | BCE | 0.12 | 0.07 | 40.23 | 42.27 |
| $\checkmark$ | FL | 0.29 | 0.19 | 38.58 | 53.12 |
| $\checkmark$ | BCE+HNM | **0.31** | **0.22** | **47.05** | **55.89** |

localization accuracy. Since the query objects in the videos are usually small, the low-resolution inputs and feature maps may not capture their details clearly. This matches our intuition that accurate localization requires relatively high-resolution features.

**Query-Frame Correspondence.** We study different methods for establishing the correspondence between the query and the frame. Specifically, we compare against a model variant that aggregates the spatial information from the query using convolutions. In detail, we apply downsample convolutions on the query features to obtain a 1-D feature vector (with $1 \times 1$ spatial resolution). It then tiles this feature to obtain a $H \times W$ feature map and concatenates it with the frame features. Finally, we use convolutions to obtain the query-to-frame correspondence features. Importantly, this does not use our Spatial TX module for establishing this correspondence. As shown in Table 3, the performance using this naive fusion method is much worse (rows 1 vs. 2), highlighting the value of establishing correspondences using our spatial transformer. We also experiment with using more transformer layers in our spatial TX module (rows 2 vs. 3 in Table 3), but this provides minimal gains, likely due to the strength of the image backbone features.

**Spatio-Temporal Transformer Designs.** Next, we study the role of using local temporal windows in our spatio-temporal TX block in Table 4. We experiment with a global window, and local windows of sizes $\{5, 7, 9\}$, with 5 being our default choice. Using a global window (row 1) significantly reduces the performance since the object's location, pose and appearance change dramatically over time in egocentric videos, making it hard to establish long-range dependencies. Instead, focusing on the local temporal window with several consecutive frames makes the feature propagation easier, establishing useful short-horizon dependencies. Our experiments show that the a window size 5 works best, where 5 consecutive frames span 1 second for the 5 FPS videos.

**Prediction head and Loss Function.** Finally, we study the design of using anchors in the prediction head and the loss functions used to supervise the anchor probabilities in Table 5. Not using anchors (row 1) performs worse than our models which use both anchors and advanced losses. Our model with anchors performs poorly with the BCE loss due to the data imbalance between positive and negative anchor boxes (row 2). The reason is that even positive frames will introduce hundreds of negative anchors, which makes the anchor-level ratio between negatives and positives much higher than the ratio in frame-level. Using focal loss or hard negative mining improves significantly over using only the BCE loss by accounting for the data imbalance (rows 3 and 4 vs. row 2). We observed that hard-negative mining (HNM) works better than focal loss (rows 3 vs. 4). Unlike HNM that only focuses on hard negatives, focal loss additionally focuses on hard positives, where the object is annotated in the training data but may be difficult to recognize due to effects like severe occlusions (e.g., only 10% of the object is visible). Therefore, these hard positives may provide harmful training signals since they are not reliably distinguishable from negatives.

**Backbone Choice.** We also ablate the backbone choice by replacing the DINOv2 encoder with a CLIP-pretrained ViT [39]. Note that the highest resolution that CLIP supports is $336 \times 336$. Experiments demonstrate with 0.28 tAP$_{25}$, 0.22 stAP$_{0.20}$, 44.67 rec% and 53.80 Succ. The results are comparable with using DINOv2 backbone, showing the versatility of VQLoC to the backbone choice.

Table 6: Model performance on different scales of objects in the videos. *s* and *l* stands for small (1941 samples) and large object (2336 samples), respectively.

| Method | Scale | $tAP_{25}$ | $stAP_{25}$ | rec% | Succ. |
|---|---|---|---|---|---|
| CocoFormer [53] | *s* | 0.206 | 0.127 | 40.804 | 32.583 |
| VQLoC | *s* | **0.213** | **0.138** | **44.719** | **33.738** |
| CocoFormer [53] | *l* | 0.338 | 0.271 | 56.164 | 40.737 |
| VQLoC | *l* | **0.454** | **0.387** | **67.680** | **53.635** |

## 5    Conclusion

We propose VQLoC, a single-stage and end-to-end trainable method for the Visual Query Localization (VQL) problem in long-horizon videos. VQLoC first builds a holistic understanding of the query-video relationship, and leverages this understanding to localize the query in a single shot. Our key insight lies in how we establish the spatio-temporal correspondence between the query and video features. Specifically, we build the query-to-frame correspondence for each video frame and then propagate these over time to nearby frames by leveraging the temporal smoothness of videos. Compared with prior stage-wise methods, VQLoC not only demonstrates better spatio-temporal localization accuracy but also improves the inference speed significantly by avoiding explicit comparison between region proposals and the query. VQLoC also achieves the top performance on the Ego4D VQ2D challenge leaderboard. In future work, we plan to develop systems to perform in-depth object-level understanding based on our model's VQL predictions.

**Limitation.**    Similar to prior works, VQLoC is trained in a supervised manner, requiring a large number of annotations for training. Besides, the hyperparameters, i.e. local window length, might need to be tuned for training or inference on other datasets with different FPS.

**Acknowledgement.**    UT Austin is supported in part by the IFML NSF AI Institute. Kristen Grauman is paid as a research scientist by Meta.

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

# Appendices

## A   Broader Impact

Visual Query Localization (VQL) has broad benefits to AR and robotics applications. For example, VQL is the basis for developing super-human episodic memory systems that can retrieve information from past human experiences. It has the potential to assist and strengthen the long-term reasoning capability of humans. Furthermore, it can be vital for embodied agents/robots to learn from past experiences or perform relevant tasks with long-term dependency.

## B   Model Details

**Model Architecture.**   We provide the workflow of our model as follows. The details of each block, i.e. Encoder, Spatial Transformer, Spatio-Temporal Transformer, and Prediction Heads are included in the main text. Specifically, $T$ is the length of a clip, and $N$ is the number of anchors. We use $T = 30$ and $N = 8 \cdot 8 \cdot 12 = 768$. We provide the code of the model class in the supplementary.

| Stage | Configuration | Output |
|---|---|---|
| 0 | Video Frames | $T \times 448 \times 448 \times 3$ |
| 0 | Visual Query | $448 \times 448 \times 3$ |
| **Encoder** | | |
| 1 | Frame features | $T \times 32 \times 32 \times 256$ |
| 1 | Query features | $32 \times 32 \times 256$ |
| **Spatial Transformer** | | |
| 2 | Updated frame features | $T \times 32 \times 32 \times 256$ |
| **Spatio-Temporal Transformer** | | |
| 3 | Downsampled frame features | $T \times 8 \times 8 \times 256$ |
| 3 | Propagated frame features | $T \times 8 \times 8 \times 256$ |
| **Prediction Heads** | | |
| 4 | Anchor refinement | $T \times N \times 4$ |
| 4 | Anchor scores | $T \times N$ |
| **If inference** | | |
| 5 | Top-1 anchor | $T \times 4$ |
| 5 | Top-1 score | $T$ |

Table 7: Model workflow.

**Inference Details.**   We provide more inference details here. For the predicted object occurrence scores on all frames, we first smooth the scores with a median filter with kernel size 5. Then we apply peak detection on the smoothed scores. We find the peak with the highest score $s$ and use $0.8 \cdot s$ as the threshold to filter non-confident peaks. We then select the response track that corresponds to the last peak as the results. To find the start and end time steps of the last response track, we threshold the occurrence scores with the value $0.7 \cdot s_p$, where $s_p$ is the score of the last peak.

## C   More Results

**Visualization.**   Please see our supp video for more visualization results of the retrieved response track with bounding boxes. Moreover, we provide the visualization of feature affinity between the query and frame. As shown below, the query point (in blue) has high feature similarity with the same object in the frame. We note that we use the pixel in the visual query as the source feature in this visualization, while we use the pixel in the target frame as the source feature in our spatial transformer. The reason is that if we visualize the feature similarity in the latter manner, the response will cover the entire visual query, which is not informative.

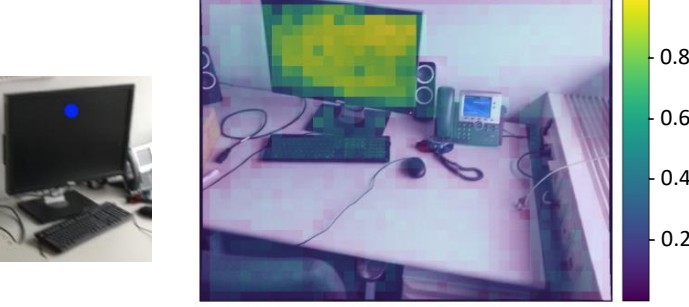

Figure 6: Visualization of feature similarity between the query and frame. The query point (in blue) has high feature similarity with the target object in the frame.

**Leaderboard Result.** Our model VQLoC achieved state-of-the-art performance on the public leaderboard of Ego4D VQ2D benchmark.

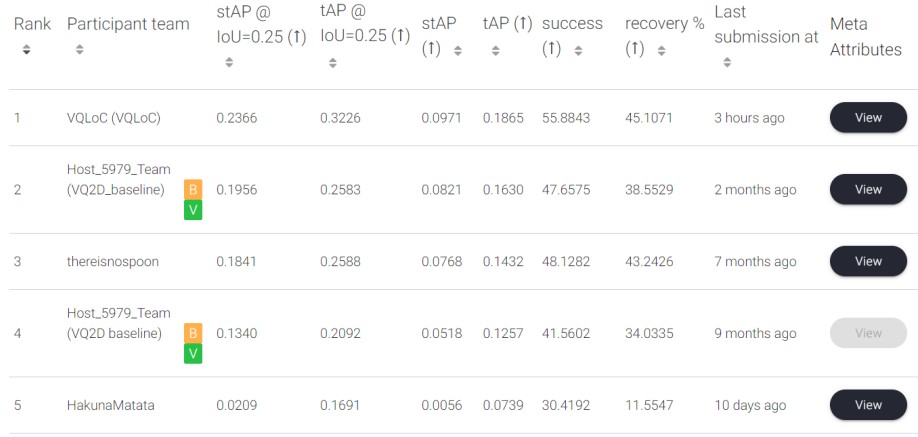

| Rank | Participant team | stAP @ IoU=0.25 (↑) | tAP @ IoU=0.25 (↑) | stAP (↑) | tAP (↑) | success (↑) | recovery % (↑) | Last submission at | Meta Attributes |
|---|---|---|---|---|---|---|---|---|---|
| 1 | VQLoC (VQLoC) | 0.2366 | 0.3226 | 0.0971 | 0.1865 | 55.8843 | 45.1071 | 3 hours ago | View |
| 2 | Host_5979_Team (VQ2D_baseline) B V | 0.1956 | 0.2583 | 0.0821 | 0.1630 | 47.6575 | 38.5529 | 2 months ago | View |
| 3 | thereisnospoon | 0.1841 | 0.2588 | 0.0768 | 0.1432 | 48.1282 | 43.2426 | 7 months ago | View |
| 4 | Host_5979_Team (VQ2D baseline) B V | 0.1340 | 0.2092 | 0.0518 | 0.1257 | 41.5602 | 34.0335 | 9 months ago | View |
| 5 | HakunaMatata | 0.0209 | 0.1691 | 0.0056 | 0.0739 | 30.4192 | 11.5547 | 10 days ago | View |

Figure 7: Ego4D VQ2D benchmark leaderboard at the time of submission.

**Error Bar.** As shown in Table 8, we provide the average value and the performance variance by running the model 3 times. The results demonstrate that VQLoC consistently achieves reasonable performance.

Table 8: Model performance and variances with multiple runs.

| Seed | $tAP_{25}$ | $stAP_{25}$ | rec% | Succ. |
|---|---|---|---|---|
| 42 | 0.31 | 0.22 | 47.05 | 55.89 |
| 0 | 0.30 | 0.23 | 45.64 | 57.32 |
| 1 | 0.31 | 0.22 | 47.73 | 55.42 |
| mean | 0.31 | 0.22 | 46.81 | 56.21 |
| var | 2e-5 | 2e-5 | 0.76 | 0.65 |

