# OpenReview forum: "Single-Stage Visual Query Localization in Egocentric Videos"
_NeurIPS.cc/2023/Conference — NeurIPS 2023 poster_

### Official Review · Reviewer_4CTN · 2023-07-03

**Soundness:** 3 good
**Presentation:** 3 good
**Contribution:** 3 good
**Rating:** 7
**Confidence:** 5

**Summary:**

The paper proposes VQLoC, an end-to-end trainable framework for Visual Query Localization (VQL) on long-form egocentric videos. Compared with Ego4D's multi-stage approaches, VQLoC proposes a single-stage process that efficiently localizes visually specified objects. It establishes both query-to-frame and frame-to-frame relationships for spatial-temporal localization. The proposed method achieves new SOTA performance on the public leaderboard with a notable improvement.

**Strengths:**

VQLoC can streamline the VQ2D localization process. The method is end-to-end trainable and achieves faster inference speed, as shown in Figure 4.

The results are verified on the public server. Furthermore, from the quantitative ablation study, it seems like the boost comes from the model design rather than parameter tuning or training tricks, which is good.

The related work covers the field well. And the authors made comparisons to previous baselines in terms of size, speed, and performance.

**Weaknesses:**

My major concern is the completeness of the ablation study. This paper can be stronger if more ablations can be done.

For the spatio-temporal transformer, Tab. 4 only shows the window size = 5 gives the best performance among the numbers greater than 5. It's not convincing 5 is the best choice. How about a window size smaller than 5? It's possible that if you don't do any frame-to-frame correspondence, i.e., a window size of 1/0 gives even better performance.

 Also, there are multiple thresholds in the methodology. Are the results sensitive to the selection of those hyperparameters? For example,  the peak response plot in Figure 5 looks noisy, so it's not sure whether an empirical choice of φ makes sense. The same also applies to the anchor box threshold theta.


**Questions:**

(P6 L207) Why not trim complete negative video clips to augment the data for the training?

(P6 L228) Are the top K anchors selected from the entire video clip n' and mixed with anchors from n? I need clarification here since the loss is frame level, but the text is about video level.

Ego4D Episodic Memory also has similar challenges like VQ3D, NLQ, and MQ. Is it possible to adopt similar strategies for these tasks by changing the query encoder and predictor? It will be quite interesting to develop a unified framework for video query tasks.

**Limitations:**

I don't think the authors can provide more ablations for a large benchmark like VQ2D within the rebuttal date. Overall, this work has pushed VQ2D task forward with a remarkable step. However, more complete ablations will make this paper stronger in the future.

---

> ### Author Rebuttal · Authors · 2023-08-09
>
> We appreciate your detailed and helpful suggestions, and we are glad to see the highly positive comments. We will address your questions below.
>
> 1. **[Local window size ablation]** We experiment with smaller window sizes 3 and 1, where 1 means there is no temporal reasoning within the model. As shown below, we observe that window size 5 is better than size 3. Moreover, without temporal reasoning, the performance drops dramatically, showing the effectiveness of the spatial-temporal transformer module we proposed.
>
> | Window Size | tAP | stAP | rec% | succ |
> |:---------:|:---------:|:---------:|:---------:|:---------:|
> | Size 5 | **0.31** | **0.22** | **47.05** | **55.89** |
> | Size 3 | 0.27 | 0.17 | 42.26 | 48.14 |
> | Size 1 | 0.17 | 0.11 | 33.56 | 42.83 |
>
> ---
>
> 2. **[Selection of hyperparameters - anchor assignment threshold]** Among the other hyperparameters,  the anchor box threshold influences the training a lot. The reason is that it is highly related to the assignment of positive and negative anchors. As the objects are usually **small** on egocentric videos (1/7 of the image resolution on average), anchor box IoU threshold 0.5 (commonly used in detection papers) does not work well since it is harder for the anchor boxes to intersect with small objects with such a high IoU. This is the reason for picking a small IoU of 0.2. We will discuss it in the later version.
> ---
>
> 3. **["Noisy" temporal plot in Fig. 5]** The peak response in Fig. 5 looks “noisy” as it shows the results on very long videos, e.g. a video of 1000 frames. Note: VQL aims to identify the ‘*latest*’ appearance track of the query object, and the ground-truth is annotated accordingly (and ignores earlier appearances of the visual query in the video). Thus, there is only a single peak in the visualized ground-truth. However, the query object itself **may appear multiple times earlier** in the video, and our model will attempt to identify all of them (not just the latest) — these correspond to the other peaks and *is the expected and correct behavior* of the model. To better demonstrate the point, we include additional visualization (in the rebuttal pdf), which shows the all identified peaks. And the results indicate that our method can identify the query object consistently.
> ---
>
> 4. **[Selection of hyperparameters - post-processing threshold]** We select the threshold based on the validation performance following the approach from the baseline methods (SiamRCNN and CoCoFormer). We agree that this is a limiting factor and can be improved in the future, but this is orthogonal to our core contributions. Moreover, our results demonstrate that this threshold determined on the validation set can generalize to the test set without a performance gap (see Table 1).
> ---
>
> 5. **[Use negative clips]** We generally believe trimming negative clips can be a promising technique to avoid false positives.  However, this is currently not possible on the Ego4D VQL dataset since *it annotates only the latest object appearance* (i.e., to answer “where did I *last* see [query]?”), while ignoring the earlier appearance of the object in the video. This means that the clips before the ground-truth response track may not all be negatives.
> ---
>
> 6. **[Anchor selection]** The loss $L_{img}$ (L210) is computed per anchor of the image, not over an image as a whole. We will better clarify it in the next version. The top K hard negative anchors are chosen from frames in video $n$ as well as frames from every other video $n’$ (see L229-231).
> ---
>
> 7. **[Unified framework]** Yes, we completely agree that these egocentric vision tasks can share similar challenges, caused by the characteristics of egocentric videos, and it is likely that these tasks have synergy. For example, NLQ and MQ require similar localization capabilities. VQ3D requires both 2D and 3D understanding of objects and scenes. Joint training with multiple tasks might improve the performance of each task. The key to the unified framework is how we handle inputs with different modalities and propose a jointly trainable end-to-end approach. It is also necessary to handle the unique properties of each task, e.g. camera pose estimation is the bottleneck of VQ3D [1]. This is an interesting and promising research direction. However, note that this is orthogonal to our core contribution of efficiently localizing a specific query object instance within the video.
>
> [1] Mai, Jinjie et al. “Estimating more camera poses for ego-centric videos is essential for VQ3D.” ArXiv 2022.

---

> > ### Comment · Reviewer_4CTN · 2023-08-19
> > **Response to the rebuttal**
> >
> > Thanks to the authors for the detailed explanation and additional info on the rebuttal. The table on local window size ablation is helpful to me. A unified framework for other tasks like NLQ, VQ3D, MQ, etc, could be interesting following this work. The authors have addressed my concerns and questions I raised. I have no further questions at this point.

---

> > > ### Author Response · Authors · 2023-08-19
> > >
> > > Thank you for your response. We are glad to hear that the rebuttal addressed your concerns.

---

### Official Review · Reviewer_8x9A · 2023-07-04

**Soundness:** 3 good
**Presentation:** 3 good
**Contribution:** 3 good
**Rating:** 6
**Confidence:** 3

**Summary:**

The author proposes a new single-stage end-to-end method for visual query localization. The proposed method builds a holistic understanding of query-video relationship, then spatial-temporal localization is performed. The proposed achieves major performance and speed gain over previous methods on major benchmark datasets.

**Strengths:**

In general, the proposed method is simple and performs well on the major egocentric Ego4D dataset. The proposed method utilizes simple pipelines with cross attention/Transformer, however, achieves state-of-the-art results. In general, I am satisfied with the paper.

**Weaknesses:**

However, I have some concerns about the paper.

1. The author only uses the Ego4D dataset as the benchmark for egocentric dataset. Since it is a new task, it is OK to only have few baselines/datasets cause it is a brand new task. However, It would be better if the author's method can be applied to other tasks/sub-tasks like visual tracking datasets (or modify the tracking datasets

2. More experiment details should be reported. Because it is an end-to-end framework. Though the author reports the FPS of the model, I think the author could report the more detailed metric like FPS/GPU memory. I think the proposed method could have larger GPU memory consumption/larger FLOPs while with fast execution time.

**Questions:**

Please mainly see the weaknesses section for details.

**Limitations:**

I think the author has adequately addressed the limitations.

---

> ### Author Rebuttal · Authors · 2023-08-09
>
> We appreciate your helpful feedback, and we will address your concerns as follows.
>
> 1. **[Benchmark]** Thank you for your valuable suggestion. In this paper, we focus on **the special properties of the visual query localization task on egocentric videos** (L33-43), e.g. drastic head motion, large variation between the query object and its appearances in the video, open-set queries, and long videos. Visual tracking, on the other hand, is a different task compared with VQL. Specifically, visual tracking usually requires the model to work on shorter videos with slower camera viewpoint changes, as well as small differences between the visual query and their future appearance in the video. Thus, your idea to apply or adapt our method to visual tracking datasets is intriguing, but we are unaware of appropriate datasets that satisfy the above properties. To the best of our knowledge, Ego4D is the only dataset that has these properties to test on. We will add these considerations to our discussion to highlight potential applications and adaptations.
> ---
>
> 2. **[More details]** Thank you for your suggestions to provide the FPS/GPU memory. In detail, our method observes 2 FPS/GPU per GB. We note that it's difficult to compute this FPS/GPU metric for the baselines, as their GPU usage is different for multiple stages of the baselines, i.e. detection, query comparison, and tracking. For an apples-to-apples comparison, we verified that the current FPS is evaluated with similar “*maximum*” GPU usage between our method and the baselines. Moreover, our method is *more parallelizable* than the baselines. If we are provided with larger GPU memory, our FPS will also increase accordingly for a given (video, query) pair.  This is not the case for the baselines as some stages like tracking cannot be parallelized (i.e., tracking has to happen sequentially from one frame to the next).

---

> > ### Comment · Reviewer_8x9A · 2023-08-15
> >
> > Thank the author for providing the rebuttal. I think your comment has addressed my concerns, thus I will keep my weak accept rating.

---

> > > ### Author Response · Authors · 2023-08-16
> > >
> > > Thanks for your response! It's great to hear that our rebuttal has addressed your concerns.

---

### Official Review · Reviewer_YaTy · 2023-07-04

**Soundness:** 4 excellent
**Presentation:** 4 excellent
**Contribution:** 3 good
**Rating:** 7
**Confidence:** 4

**Summary:**

The paper introduces a new approach to address the visual query localization problem in egocentric videos. The major contribution is that the proposed method, a single-stage model, simplifies the previous multi-stage frameworks and eliminates the need for off-the-shelf object detectors, tracking, and similar components by utilizing an end-to-end trainable model. Notably, the proposed method has achieved the top position in the Ego4D VQ2D challenge as of the submission date, demonstrating its effectiveness. Besides, the method demonstrates a considerable enhancement in inference speed, further highlighting its advantages.

**Strengths:**

1. The paper introduces a novel and technically sound method that formulates the visual query localization task as a unified framework. By taking a video and a template query image as input, the proposed method can directly produce the track of bounding boxes, eliminating the need for off-the-shelf object detectors and tracking methods. This is achieved by decoupling the task into spatial reasoning (finding per-frame responses) and temporal reasoning (consultation within the clip), followed by a prediction head (box&score) upon the learned features. In summary, the idea behind the method is both original and rational.

2. The proposed method demonstrates notable computational efficiency, a crucial aspect for real-world applications. By simplifying previous multi-stage frameworks into a one-stage process, the method achieves a remarkable 10x improvement in speed while maintaining a satisfactory level of performance. This advancement represents a significant stride towards practical applications.

3. The paper is well-written. Readers with a relevant background will have no difficulty following and comprehending the content.

4. The paper presents solid experimental results, achieving the top position on the Ego4D VQ2D challenge leaderboard at the date of submission.

**Weaknesses:**

1. L12-L13: "Our experiments demonstrate that our approach outperforms prior VQL methods 12 by 20% accuracy while obtaining a 10× improvement in inference speed". Without delving into the entire paper, a glance at Table 1 and Figure 4 might cause confusion regarding the reported results, as the speed does not appear to be 10x faster than STARK [51] (if my understanding is correct, this is an adapted tracking approach upon the proposed framework). It would be beneficial to include some explanation and details in the caption to prevent such confusion.

2. It is unclear how the number of frames (T=30) per clip is determined? Is this choice based on the limit of computational resources or driven by experimental performance considerations. If the latter, it would be helpful to know if any ablation studies have been conducted to investigate its aspect.

3. Discussion on the long videos. The input is fixed to a maximum of 30 frames (~6 seconds with a frame rate of 5fps). To handle longer videos, the proposed solution is to chunk the untrimmed video into fixed-length clips. In this case, is there an additional step to ensure smooth predictions between these clips, such as incorporating tracking between them? Alternatively, another solution to address long videos is to use a streaming mode. Suppose we have processed the past 30 frames $\{ t_{i-29}, …, t_{i-1}, t_i \}$, we would only need to process the new incoming frame $t_{i+1}$. Spatial transformer naturally lends itself to this scenario since it is computed at frame level. The spatio-temporal transformer might be more complex. But it operates within a bounded window, which suggests that it might not require significant re-computation or adjustments for the streaming mode. It would be valuable to have the authors' insights and comments on this matter.

4. L177-L193: The symbol $w$ is used ambiguously, representing both the spatial width and the temporal window.

**Questions:**

Please refer to the previous section.

**Limitations:**

The authors addressed the limitations.

---

> ### Author Rebuttal · Authors · 2023-08-09
>
> We highly appreciate your insightful comments and your suggestions on the paper details. We will address your questions as follows.
>
> 1. **[Speed comparison]** We apologize for the confusion. The speed improvement was not in comparison to STARK but the prior state-of-the-art VQL methods (SiamRCNN / NFM / CoCoFormer). We will clarify this in our writing.
> ---
>
> 2. **[Clip length]** The clip length (T=30) was chosen based on the computational resource limits and the average response track length (~15 frames). This selection offers a *balanced ratio* between positives and negatives within each clip during training, ensuring steady training. Since we use local-windowed attention to handle fast head motion in egocentric videos, extending the clip length beyond T=30 would not theoretically enhance performance. For example, with a window size of 5 and 3 layers of local-windowed attention, frames more than approximately 7 frames away will not be attended to. We will clarify this in the later version.
> ---
>
> 3. **[Long videos]** We appreciate the insightful comments. As we use local-windowed attention to perform spatial-temporal reasoning, the smoothness of predictions within the several consecutive frames is implicitly learned. As you suggested, it is still possible that the prediction is not smooth enough between the two clips, e.g. the last frame of a clip, and the first frame of the following clip. The streaming mode you suggested is interesting and may further improve the smoothness at the intersection of clips. Further, the local-windowed attention can be used in the streaming mode with slight modification. It would be interesting to explore this in future work. Besides, we think inference in a streaming mode is especially important for egocentric vision, as it usually requires real-time inference. We will add more discussion in the later version.
> ---
>
> 4. **[Writing]** Thanks for pointing this out. We will fix it.

---

> > ### Comment · Reviewer_YaTy · 2023-08-15
> > **Re: Rebuttal by Authors**
> >
> > I appreciate your response. I'm pleased to hear that you agree with my points (especially, intend to incorporate the steaming function into your future work) and their inclusion will undoubtedly enhance the clarity and quality of the work.

---

> > > ### Author Response · Authors · 2023-08-16
> > >
> > > Thanks for your response! It's great to have a discussion on future work directions with you.

---

### Official Review · Reviewer_bR3g · 2023-07-06

**Soundness:** 3 good
**Presentation:** 3 good
**Contribution:** 2 fair
**Rating:** 4
**Confidence:** 5

**Summary:**

This paper tackles the problem of query localization of visually specified objects in egocentric human-activity videos. The main technical contribution for the solution is to design a single-stage transformer-based architecture to model the query-to-frame correspondence matching and frame-to-frame correspondence propagation. Ablation studies verify the effectiveness of the proposed components and comparisons show the superiority of the method in performance and speed with respect to prior works.

**Strengths:**

1) The network is single-stage and end-to-end trainable, which alleviates the issues in existing multi-stage approaches that different stages are separate from each other for conducting non-differentiable stage-wise predictions.
2) Model designs, including the frame-wise spatial transformer and the subsequent spatio-temporal transformer, are technically sound and easy to follow.
3) Good performance is achieved by the proposed method and speed-accuracy trade-off is also taken into consideration.


**Weaknesses:**

1)	The technical contributions in terms of the model designs are short of novelty, and most of the building blocks can be viewed as simple adaptations from existing techniques used in methods for similar tasks (e.g., visual tracking, temporal sentence grounding, spatio-temporal video grounding, etc., can refer to STARK[1], LGI[2], STCAT[3], QD-DETR[4]). Apart from that, the unique challenges in the task of VQL, such as the egocentric characteristics presented in the videos, are less considered and are scarcely reflected by the model designs.
2) A powerful pre-trained model DINOv2 is adopted as the visual backbone, which plays a crucial role for ensuring the success of the query-to-frame correspondence matching. However, no related discussions are included in the manuscript. What if the method gets rid of (by replacing it with a weaker but common one) the pre-trained DINOv2 model? Would it simply fail or drop significantly in performance? And it’s unclear how much gains are brought by the proposed architecture except for the utilization of a stronger feature backbone.
3) According to the ablation studies, the proposed model heavily relies on a high input resolution to achieve decent localization results. This could be a negative factor hindering the VQL model from scaling up to a larger amount of training data.
4) Some literal expressions, such as the writing of Section 3.3 could be reorganized to make it clearer.
5) As shown by the temporal plot of predicted object-occurrence probabilities in Figure 5, there are still a lot of spurious responses with large magnitude (some are even larger than responses within GT) outside the target temporal region, which implies the current model’s deficiency in accurate temporal localization to some extent.

[1] Bin Yan, et al. "Learning spatio-temporal transformer for visual tracking." In ICCV, 2021.
[2] Mun, Jonghwan, et al. "Local-global video-text interactions for temporal grounding." In CVPR, 2020.
[3] Jin, Yang, et al. "Embracing consistency: A one-stage approach for spatio-temporal video grounding." In NeurIPS, 2022.
[4] Moon, WonJun, et al. "Query-dependent video representation for moment retrieval and highlight detection." In CVPR, 2023.


**Questions:**

None

**Limitations:**

YES

---

> ### Author Rebuttal · Authors · 2023-08-09
>
> We appreciate your valuable comments. We will address your concerns as follows.
>
> 1. **[Novelty and related work]** We respectfully disagree with the comment. Our method is not a simple adaptation of the existing methods. Its novelty is *appreciated by other reviewers*, and we reiterate our contributions here. We *take the challenges and unique characteristics of egocentric videos into consideration* and propose three ways to address them: i) The **spatial transformer** finds the correspondence between the frame-query to identify regions that are similar to the query. This is designed to **handle the large appearance variation** of the query object with its appearance in the egocentric video; ii) The **spatial-temporal transformer** performs temporal reasoning within locally windowed consecutive frames to propagate spatial correspondences identified earlier. This local window is necessary to **handle the fast head motion** of the egocentric videos. iii) The **anchor-based bounding box prediction with hard negative mining** during training is designed to **handle the small scale and rapid appearance of the objects** in the egocentric videos. These egocentric characteristics and proposed techniques differentiate our task/model from previous visual tracking/grounding tasks and solutions. And our performance without these components drops 35%, 80%, and 40%, respectively. In contrast, for frame-query feature fusion, STARK and STCAT use concatenation; LQI uses Hadamard product. For temporal reasoning, STARK updates the template; LGI, STCAT, and QD-DETR perform reasoning on the entire clip. For bounding box (or target frame) prediction, all mentioned works use non-anchor-based loss without hard negative mining. Since STARK is a SOTA tracker and can be directly used to localize the visual queries, it is tested as a baseline and demonstrates degenerated performance on the VQL task (Table 1). We will add more discussion and cite these papers in the future version as you suggested.
> ---
>
> 2. **[Backbone]** Good question. Our method **does not fail** when we replace DINOv2 with a common CLIP backbone, which is widely used in video-related [1] or non-video [2] tasks.  The performance with the CLIP backbone is considerably better than the baseline models and demonstrates that our model still works with alternate backbones. Note that large-scale pre-training is a valuable component of most existing methods [3] (whether it is CLIP or DINOv2), and it is not surprising that it plays an important role here. However, the choice of DINOv2 as the particular backbone for this task is also a part of our contribution (see L155-157). As contrastive-learning-based backbones, including DINOv2, demonstrate good *semantic correspondence properties* [4], it is useful for performing the visual query task, especially for handling *large visual differences* between the query and its appearance in the videos. Besides, the backbone is not the only important aspect of our model. Even with a strong DINOv2 backbone, we demonstrate in our ablations that the performance deteriorates significantly without our other proposed components, as illustrated in the reply to your first question. In short, each component of our model is designed to work well for the VQL task, and i) the performance cannot be solely attributed to the backbone and ii) our CLIP test demonstrates some versatility to the backbone.
>
> | Backbone | Resolution | tAP | stAP | rec% | succ |
> |:-------:|:-------:|:-------:|:-------:|:-------:|:-------:|
> |DINOv2|448| 0.31 | 0.22 | 47.05 | 55.89|
> | CLIP | 336 | 0.28 | 0.20 | 44.67 | 53.80 |
>
> *Note that the highest resolution that CLIP supports is 336.
>
> [1] Lin, Ziyi et al. “Frozen CLIP Models are Efficient Video Learners.” ECCV 2022.
> [2] Shen, Sheng, et al. "How much can clip benefit vision-and-language tasks?." ICLR 2022.
> [3] Wang, Yi et al. “InternVideo: General Video Foundation Models via Generative and Discriminative Learning.” ArXiv 2022.
> [4] Hu, Yingdong et al. “Semantic-Aware Fine-Grained Correspondence.” ECCV 2022.
>
> ---
>
> 3. **[Input resolution]** We would like to clarify that our model is **not dependent on high-resolution inputs**. For comparison, we note that baselines, i.e. SiamRCNN, NFM, and CocoFormer, work on original video resolution, which is 1200 pixels. In contrast, **our method works on downsized resolution** of 448 pixels, yet outperforms the baselines. This indicates that our method is effective even though it uses lower resolution and is not over-reliant on high-resolution inputs. Besides, the performance drop with using a smaller resolution (i.e. 224) is expected and is not unique to our method [5], as the resolution of visual features will also be halved. Specifically, with 224 resolution, the average object bounding box size is about 35 pixels (i.e., 15% of the image length), which is challenging to identify even for humans. We will better emphasize this point in our revised version.
>
> [5] Bertasius, Gedas et al. “Is Space-Time Attention All You Need for Video Understanding?” ICML 2021.
>
> ---
>
> 4. **[Writing]** We will revise Sec.3.3 to improve clarity.
> ---
>
> 5. **[Spurious responses in Fig.5]** It is crucial to note that VQL aims to identify the ‘*latest*’ appearance track of the query object. This means that the ground-truth is annotated only with the latest appearance of the query object and ignores all prior occurrences in the video. Thus, there is only a single peak in the visualized ground-truth. However, since the query object itself may appear multiple times earlier in the video, our model will attempt to identify all of them (not just the latest) — these correspond to the other peaks (or “spurious responses” as noted) and is *the expected and correct behavior* of the model. To better demonstrate the point, we include additional visualization (in the rebuttal pdf), which shows all identified peaks. The results indicate that our method can identify the query object consistently.

---

> > ### Comment · Reviewer_bR3g · 2023-08-16
> >
> > Thank the authors for providing the rebuttal. Your comments have addressed my concerns.

---

> > > ### Author Response · Authors · 2023-08-16
> > >
> > > Thank you for the response. We appreciate your valuable review and we will incorporate your suggestions to the later version accordingly.

---

> > > ### Author Response · Authors · 2023-08-18
> > >
> > > Dear reviewer, thank you for acknowledging that the rebuttal addressed your concerns. I noticed the rating remains unchanged; are there any further aspects I should address? Your feedback is highly valued.

---

### Official Review · Reviewer_59Vz · 2023-07-07

**Soundness:** 3 good
**Presentation:** 4 excellent
**Contribution:** 3 good
**Rating:** 7
**Confidence:** 5

**Summary:**

The paper proposes VQLoC, an end-to-end method for egocentric visual query localization based on a holistic understanding of the query-video relationship. The key component is a spatio-temporal transformer, which can effectively model the relationships between the query and frames. The presented approach can perform single-stage inference and achieve the winning entry on the leaderboard. Moreover, the method produces a set of models to balance the speed and balance, while the top-performing one is more advanced from both perspectives than the state-of-the-art. Moreover, the visualization shows that VQLoC is excellent at the visual query 2D detection task and can even deal with challenging cases such as clutter, occlusions, and motion blur.

**Strengths:**

1. This paper positively impacts the egocentric community, especially in episodic memory tasks. Unlike Moment Query and Language Query, the visual query task is more complicated in spatial-temporal localization, and the existing solutions are redundant. The paper provides a single-stage solution, and the chosen model is more accurate and efficient than the state-of-the-art. This will encourage more researchers to work on the valued research problem.

2. The proposed architecture for the spatio-temporal transformer is able to effectively model the relationships between the query to each frame as well as between the frames. Therefore, the model can leverage rich semantics during feature embedding. Locally-windowed temporal self-attention is applied to improve the model efficiency without losing too much information.

3. Code and the architecture details are attached to the paper, which helps people to re-implement the code. Also, the high-quality video in the supplementary material makes the paper clear and easy to follow.

**Weaknesses:**

1. My main concern is that although the paper aims to solve the Visual Query Localization task, all the experiments are conducted in the VQ2D setting. According to the definition of the Ego4D [13] paper, visual query localization can be done in the video domain (VQ2D) or the real-world coordinate (VQ3D). Therefore, it is not precise to only validate this method on VQ2D, and evaluating the proposed method on the 3D setup is highly recommended. Otherwise, the paper should revise the task as visual queries 2D localization.

2. Further experiments could be conducted to improve the model further. For instance, the authors find the local windows
of size 5 works the best in the set {5, 7, 9}, then they should experiment with smaller windows for the local optima of their hyper-parameters.

**Questions:**

1. In the ablation study L299, using high-resolution frames (448 × 448) leads to way better results than the low-resolution ones (224 × 224). Is it possible to further increase the resolution for better performance, even if we can slightly sacrifice the clip length?

2. It seems that the prediction head only gives framewise bounding box locations and confidence scores, but the VQ2D task requires a response track of the query object. Is there any mechanics in the head to make the predicted boxes consistent with the predictions from the temporal neighboring frames?

3. According to Table 1, the recovery ratio and success rate improved significantly, but the AP only raised a little. Is it because the single-stage detection pipeline is weak in predicting precise bounding boxes?

**Limitations:**

The limitation is already discussed in the paper.

---

> ### Author Rebuttal · Authors · 2023-08-09
>
> Thanks for the positive comments, and your acknowledgment of our potential impact to the community! We will address your questions as follows.
>
> 1. **[Weakness Q1]** Our study intentionally focuses on VQ2D due to its unique challenges, e.g. the ‘needle-in-the-haystack’ problem, and the large variation between visual queries and their appearance in the video, as we discussed in L33-43 of the submission. While VQ3D is interesting,  it poses unique challenges that *are orthogonal to* VQ2D [1], i.e. difficulties in camera pose estimation, which should be solved separately and is beyond the scope of our work. However, we expect our improvements in VQ2D to propagate to VQ3D since 2D localization is a precursor to 3D localization. We will revise the task as visual query 2D localization as you suggested to make it more clear.
>
> [1] Mai, Jinjie et al. “Estimating more camera poses for ego-centric videos is essential for VQ3D.” ArXiv 2022.
>
> ---
>
> 2. **[Weakness Q2]** We experiment with smaller window sizes 3 and 1, where 1 means there is no temporal reasoning within the model. As shown below, we observe that window size 5 works the best. Moreover, without temporal reasoning (i.e., window size = 1), the performance drops dramatically, showing the effectiveness of the spatial-temporal transformer module we proposed.
>
> | Window Size | tAP | stAP | rec% | succ |
> |:---------:|:---------:|:---------:|:---------:|:---------:|
> | Size 5 | **0.31** | **0.22** | **47.05** | **55.89** |
> | Size 3 | 0.27 | 0.17 | 42.26 | 48.14 |
> | Size 1 | 0.17 | 0.11 | 33.56 | 42.83 |
> ---
>
> 3. **[Questions 1]** While higher resolution could theoretically enhance the performance, our computation resources restrict us to the 448x448 resolution. We are currently training with batch size 3 on each GPU. If we intend to increase the resolution further, e.g. to 896x896, the batch size should be decreased to one-fourth, as the memory requirement of attention-based models grows quadratically. And in this case, we cannot even run on a batch size of 1. If there are GPUs with larger memory available, we believe it is possible to further increase the resolution. Moreover, we are running on a clip length of 30 frames. As the average response track is 3 seconds (15 frames), this clip length leads to a balanced ratio between positive and negative frames during training. Thus, decreasing the clip length in an effort to accommodate higher-resolution images may make training unbalanced, and the smaller batch size may also negatively influence the performance.
> ---
>
> 4. **[Questions 2]** Note that the prediction of each frame *is not isolated from nearby frames*. The proposed spatial-temporal transformer with local-windowed attention allows the information to be propagated across consecutive frames by establishing frame-to-frame correspondence, which implicitly makes the prediction smooth (L177-179 in the paper). Our experiments (Table 4 and our response to WQ2) and visualization in the original supplementary video demonstrate the effectiveness of using the spatial-temporal transformer to make the performance better and smooth.
> ---
>
> 5. **[Questions 3]** We note that our method demonstrates significant and consistent *relative* improvement in all metrics: 19% tAP, 16% stAP, 24% rec%, and 17% succ., when compared to the best baseline method. Among all the metrics, the stAP is the most challenging one, as it requires high precision in both temporal and spatial understanding. Thus, the absolute value of stAP is relatively low for all methods, as are the absolute differences in stAP between methods.

---

> > ### Comment · Reviewer_59Vz · 2023-08-18
> >
> > Thanks for providing the supplementary experiments and answering the extra questions. My concerns have been well addressed. Please don't forget to revise the paper accordingly.

---

> > > ### Author Response · Authors · 2023-08-18
> > >
> > > Thanks for the response. It's great to hear that our rebuttal addressed your concerns. We will revise the paper accordingly.

---

### Author Rebuttal · Authors · 2023-08-09

We appreciate the insightful feedback and detailed suggestions from the reviewers. It’s great to see the highly positive comments, including “positively impacts the egocentric community” (R-59Vz), “The paper introduces a novel and technically sound method” (R-YaTy), “is simple and performs well” (R-8x9A), and “VQLoC can streamline the VQ2D localization process” (R-4CTN). We will address the questions of each reviewer separately. Besides, we add additional visualization in the rebuttal pdf to answer the "noisy" peak problem.

---

### Decision · Program_Chairs · 2023-09-21

**Decision:**

Accept (poster)

**Comment:**

This work proposes a novel single-stage solution for visual query localization. Through the single-stage design, the proposed method achieves a 10x speedup while maintaining satisfactory performance.

In the process of rebuttal and discussion, reviewers raise multiple questions regarding 2D/3D settings,  pre-trained backbone, detailed parameters and more ablation studies. All reviewers agree that the rebuttal well addressed their concerns.

After the discussion period, this submission receives 3 “Accept”, 1 ”Weak Accept” and 1 “Borderline Accept”.

It is recommended for acceptance and the authors should take the comments into consideration to refine the paper in the camera-ready version.